# Participant perception, still a major challenge to clinical research in developing countries—A mixed methods study

**Wafaa Binti Mowlabaccus** ◉*, **Abha Jodheea-Jutton**◉

Department of Medicine, University of Mauritius, Moka, Mauritius

◉ These authors contributed equally to this work.
* wafaab945@gmail.com

## Abstract

### Background

With the introduction of the Clinical Trial Act in 2011, Mauritius has witnessed significant progress in the running of clinical trials. Our aim was to provide insights on the perception of clinical trials among Mauritian citizens and highlight areas of opportunities to address gaps in public perception and awareness.

### Population, materials and methods

A mixed study was carried out which consisted of 2 phases: a qualitative, with thematic approach followed by a quantitative study with cross-sectional design. For the qualitative study, computer literate individuals were invited to fill an unstructured, online open-ended questionnaire. Conclusions derived from the latter were used to adapt a validated questionnaire which was then distributed to 400 consented participants.

### Results

There were 23 participants who responded to the online qualitative survey, which showed poor knowledge and diverse views on clinical trials. Quantitatively up to 48% of the participants were not aware of clinical trials which included people of older age group, those from low socioeconomic status and those with low literacy level (p<0.050). Majority of respondents agreed with the value of research while a minority had poor perception related to trust in research companies and conduct of clinical trials. Respondents who had previously engaged in clinical research had better knowledge and perception compared to those who did not participate in one (Odds Ratio = 1.7).

### Conclusion

This novel study provided a foundation of how Mauritian citizens perceive clinical trials. Public awareness and educational programs can be created to address lack of awareness, the negative perceptions and knowledge gaps of clinical trials among Mauritian citizens.

**Data Availability Statement:** All raw data files are available from the Figshare database (DOI: 10.6084/m9.figshare.12198009.v2).

**Funding:** The author(s) received no specific funding for this work.

**Competing interests:** The authors have declared that no competing interests exist.

## Introduction

Clinical trials remain an essential pillar of biomedical science facilitating translational research. According to the International Clinical Trials Registry Platform (ICTRP), the global number of registered clinical trials increased from 3294 in 2004 to 23 384 in 2013 annually [1]. Essential elements in the advancement of medicine and development of new drugs include public confidence, their trust in the pharmaceutical companies, and a positive feeling towards research [2]. Public awareness and perception play an essential part in the setting up of new regulations and refining ethical standards regarding the conduct of research [3]. Numerous factors can shape the perception of the general population. These include socio-demographic status, level of education and exposure to clinical trials itself [4].

With a Gross Domestic Product of around 13.34 billion US dollars, Mauritius, a small developing island, located in the Indian Ocean, is undergoing continuous economic growth and has appealed to important domestic and Foreign Direct Investors [5]. Moreover, the establishment of a legal framework for clinical trials in 2011 boosted the interest of sponsors to conduct interventional trials in the island. The Government of Mauritius has introduced the Clinical Trial Act, adopted by the National Assembly on the 19 April 2011, to provide the legal framework to conduct clinical research. Three committees were set up; Clinical Research Regulatory Council (CRRC), Ethics Committee (EC) and a Pharmacovigilance Committee (PC). The rapidly evolving concept of clinical trials in Mauritius beckons the need to assess the knowledge and perception of participants. Since potential clinical trials participants are recruited from the general population, assessing their perception and knowledge about clinical research is highly significant, especially in a setting where demand for clinical trials patients outpace the supply [2, 6, 7]. A study conducted in this line of thought might be a realistic way to reveal the roadblocks of conducting clinical trials and subsequently help to improve the perception of the general public through awareness campaigns. It is also imperative that a benchmarking of perceptions be considered to monitor the effectiveness of these awareness campaigns. Up to date, no such studies have been carried out to evaluate this aspect of clinical trials in Mauritius.

### Aims and objectives

This study aims to evaluate the perception of the general population in an economically emerging setting, where the scope for clinical research is constantly growing.

The objectives of the study are to:

- Assess the level of knowledge and the attitudes of participants on clinical trial

- Explore the opinions of people about clinical trials and their willingness to participate

- Correlate awareness of clinical trials with socio-demographic factors

## Materials and methods

This is a mixed method study consisting of a qualitative phase that addresses the attitudes towards clinical trials, and a quantitative aspect which shows evidence of the knowledge paucity among participants.

### Ethical clearance

Ethical clearance was obtained on the 23$^{rd}$ of August 2018 from the Department of Medicine Research Ethics Committee of the University of Mauritius following submission of a research protocol.

## Study design

Phase I: For the qualitative study, an unstructured questionnaire containing five open-ended questions was distributed among literate participants aged from 20 to 70 years,. All willing participants were required to sign a consent form and fill out an online questionnaire consisting of unstructured questions. All the information obtained from the questionnaire was transcribed and coded by the two authors. Data was collected till saturation of themes was reached. A total of 23 responses were obtained. Thematic approach was then used to analyze the answers obtained. Conclusions obtained from this study helped to adapt a validated questionnaire used in India and Korea [2, 6].

Phase II: An adapted version of the Korea/Indian validated questionnaire was used to assess the beliefs of participants about clinical trials. Questions such as desired source of information, willingness to participate in a clinical trial and Mauritius benefiting from research were added for those who would not be able to answer questions related to perception and knowledge.

## Population and setting

Participants aged between 20 to 70 years were randomly chosen from the 9 districts of Mauritius. Two different supermarkets from each district were randomly chosen and every 10th adult coming out of the supermarket was approached. The participants were given a participation sheet that outlined the purpose of the study and were assured that anonymity and confidentiality will be respected. Once they consented they were asked to fill the questionnaire.

## Sample size calculation

Based on the *Population and Vital Statistics Report of the Republic of Mauritius* in 2017, the General Mauritian population aging from 20 to 65+ years is around 933 157 [8]. The single proportion sample size formula was used to calculate the sample size with a 95% confidence interval and a 5% margin of error. The minimum sample size obtained from this calculation was 364 participants.

## Statistical analysis

Data was analysed using Microsoft Excel and Statistical Package for the Social Sciences (SPSSv23.0). Socio-demographics were summarized using descriptive statistics. Categorical data were summarized in frequencies and percentages. To establish an association between sociodemographic factors and awareness of clinical trials amongst the general Mauritian population, Chi-square and Mann-Whitney U test were used. Association between education level and perception of clinical trials were evaluated using the Kruskal Wallis test. Finally, to assess the perception and knowledge, scores were calculated using defined standards, by assigning 1 point for each correct answer. A p- value less or equal to 0.050 was taken to be statistically significant.

## Results

### Subject characteristics and response rate

To get in-depth insights into the perceptions of adults on clinical trials, a total of 35 questionnaires were distributed for the qualitative study and 23 participants filled the semi-structured form. For the quantitative study, 400 questionnaires were distributed of which 364 were collected and 350 of them were considered to be properly filled. The response rate for part I of the study was 67.5% while that of part II was 87.5%. Table 1 describes the characteristics of the population for the quantitative study.

**Table 1. Characteristics of respondents.**

| Characteristics | Heard about clinical trials (N = 183) | Did not hear about clinical trials (N = 167) | Total (N = 350) | p value |
|---|---|---|---|---|
| **Age** | | | | <0.001 |
| Mean Age | 35.5±14.0 | 44.5±14.0 | 39.9±14.6 | |
| **Gender** | | | | 0.020 |
| Male | 76(41.5%) | 97(58.1%) | 173(49.4%) | |
| Female | 107(58.4%) | 70(41.9%) | 177(50.6%) | |
| **Education level** | | | | <0.001 |
| Primary level | 9(4.9%) | 66(39.5%) | 75(21.4%) | |
| Secondary level | 72(39.3%) | 45(26.9%) | 117(33.4%) | |
| BSc | 79 (43.1%) | 21(12.6%) | 100(28.6%) | |
| Masters | 23 (12.7%) | 35(21.0) | 58(16.6%) | |
| **Average income per month** | | | | 0.163 |
| Rs 0-Rs 5000 | 40(21.9%) | 26(15.6%) | 65(18.9%) | |
| Rs 5000-Rs 10 000 | 33(18.0%) | 31(18.6%) | 64(18.3%) | |
| Rs 10 000- Rs 20 000 | 66(36.1%) | 56(33.5%) | 122(34.9%) | |
| Rs20 000- Rs 50 000 | 20(10.9%) | 33(19.7%) | 53(15.1%) | |
| Greater than Rs 50 000 | 24(13.1%) | 21(12.6%) | 45(12.1%) | |
| **Economic Status** | | | | <0.001 |
| Student | 36(19.7%) | 6(3.6%) | 42(12.0%) | |
| Employed | 104(56.8%) | 56(33.5%) | 160(45.7%) | |
| Unemployed | 28(15.3%) | 74(44.3%) | 102(29.1%) | |
| Retired | 8(4.4%) | 16(9.6%) | 24(6.9%) | |
| Housewife | 7(3.8%) | 15(9.0%) | 22(6.3%) | |

An income less than 5000 Mauritian rupees ($124.30 US) per month for one adult is considered to be below the poverty line [9]. Exchange rate at time of survey was $1 US = Rs 40.25).

## Knowledge of clinical trials

On qualitative analysis, the knowledge of participants about clinical trials was limited. There were some participants who were unclear on what a clinical trial was. Out of 23 responses, there was one participant who claimed he did not know anything about the particular subject. There was a tendency to associate clinical trials with research. A participant mentioned: "*I am not quite sure but it seems to be research work*", while another one said: "*I guess it is medical experiments carried out on animals.*" However, there was a generalized vagueness with regards to the concept of clinical trials. We had one exceptionally accurate answer: "*I believe it is a research study carried out to evaluate the effectiveness of medications on different patients or participants in the research*". Although the exact meaning was unclear, there was an importance attached to 'Informed consent', which was felt to be a mandatory requirement prior to participation in a clinical trial. This was put as: "*everything must be based on participant's consent; more so at each and every stage of the research consent must be sought.*"

On the other hand, quantitatively, only 52.2% of respondents heard about clinical trials. Younger participants and female participants were more likely to have heard about clinical trials as well as participants who are employed or have higher education and employed (Table 1).

To assess knowledge, 13 True/False questions (Table 2) were asked to participants who knew about clinical trials (N = 183). Each correct response was given 1 point and each incorrect answer was given 0 point. Table 2 shows the percentages of correctly answered questions among those who heard about clinical trials (Table 2). A score less than 6 out of 13 was

**Table 2. Questions to assess knowledge among those who heard about clinical trials.**

| Knowledge questions | Percentage of correct answer |
|---|---|
| Clinical trials include filling survey. | 63.3% |
| Clinical trials include testing new drugs. | 24.2% |
| Clinical trials include testing new devices. | 53.5% |
| Healthy individuals with no significant health problems can participate in a clinical trial. | 48.1% |
| Individuals battling for life threatening disease e.g. cancer can participate in a clinical trial. | 57.2% |
| Individuals with health issues like hypertension can participate in a clinical trial. | 51.5% |
| All treatments, medical devices must be tested in clinical trials before being marketed and sold. | 51.4% |
| Pharmaceutical companies do research to cure more diseases and do so more effectively. | 69.2% |
| Pharmaceutical companies do research to find out whether new medications will be more effective and secure. | 68.1% |
| Pharmaceutical companies do research so that physicians can have new medications. | 54.6% |
| Participants receive free medical extra services besides the experimental drug such as education about their disease, nutritional evaluation and guidelines, medical equipment (e.g., glucometer), etc. | 65.3% |
| By participating, patient will help advance science treatments of diseases. | 50.3% |
| People participate in clinical trial because they will have the newest medicine available on the market. | 51.2% |

considered to be poor knowledge and a score greater or equal to 6 out of 13 was said to be good knowledge. 43.2% respondents had a good knowledge (Table 3), however participants who have participated in trials before are more likely to have good knowledge on clinical trials (Odd Ratio = 1.7).

## Perception of clinical trials

**Economic development/business.** Participants believed that the country will benefit with the running of clinical trials. They are convinced that Mauritius will boost its economy and attract more potential international companies to come and invest in the island as per quotes from participants.

"*The pharmaceutical companies may consider investing in Mauritius, thus boosting its economy.*"

"*International recognition as a research country and thus more international companies investing in Mauritius causing revenue.*"

One participant even said that by encouraging the running of clinical trials in Mauritius, more employment will be created. While it was commonly perceived that Mauritius will bene-fit from clinical trials, there were also concerns of possible problems from socio-cultural

**Table 3. Knowledge on clinical trials and participation.**

|  | Previously participated in a clinical trial (N = 57) | Never participated in a clinical trial before (N = 126) | Total (N = 183) |
|---|---|---|---|
| Good Knowledge | 30(16.4%) | 49(26.8%) | 79(43.2%) |
| Poor knowledge | 27(14.8%) | 77(42.1%) | 104(56.8%) |

groups, who are against the use of animals for research. "*No benefits. Socio-cultural organizations will bleed the government*".

## Perception of clinical trials and literacy level

Quantitatively 76% of respondents perceived that Mauritius will benefit from the running of clinical trials. The questionnaire consisted of 20 true/false/not aware questions set to assess perception of CT. All respondents who heard about CT were assessed for the level of perception using a perception score where a positive answer was given a score of 1 and incorrect/not aware answers were given a score of 0. While the average perception was 13 out of 20, a better perception score was found to be associated with a high literacy level (p value = 0.02).

## Perception of clinical trials and previous participation

We further assessed how perception differed between those who previously participated in a trial and those who did not. Perception questions were divided into 4 categories; value (V1-4, Table 4), motivation (M1-4), conduct (C1-4), and trust (T1-6) in research companies. The percentages of correctly answered questions related to perception were calculated and stratified in 2 groups; those who previously participated in a trial (N = 57) and those who did not (N = 126) (Table 4). Assessing how perception differed between these 2 groups revealed that there was no notable difference in how respondents perceived the importance of clinical trials and research (p = 0.929). However, those who have participated in clinical trials before were more motivated (p = 0.027) to participate in research and also perceived the conduct of clinical trials more positively (p = 0.040) than those who never participated before. Finally, respondents who previously participated in a clinical trial were found to have a higher perception score than those who never participated before (p value = 0.020).

## Fears and benefits

**Death and high risk venture.** From the qualitative study, concerns and fear regarding clinical trials were frequently picked up. The main concerns revolved around the side effects and safety of the medication. Some strongly claimed they would not participate in a clinical trial that could potentially harm or cause them pain. Potential life threatening side effects were also considered as a major setback in participating. A participant mentioned: "*Clinical trials may expose people to untested situations and might be harmful to the health of the candidate.*" Some participants were also unwilling to be treated as '*subjects*' and '*guinea pigs.*' Moreover, lack of knowledge about the particular topic was a deterrent factor to participation. There was also mention of "*. . .I know too little on the subject and therefore won't venture into it as it is a matter of health and on that matter "better be safe than sorry" is the ideal idiom . . .*" mirroring the lack of knowledge about clinical trials.

**Willingness to participate.** Participants indicated that they will be willing to participate in clinical trials only if they are sufficiently remunerated. One participant said: "*Yes, if fairly safe and paid.*"

Another determining factor to participate in clinical trials for some people was to improve their own health status. The participants mentioned that they would be willing to participate in clinical trials if they were suffering from a terminal disease. One participant said, "*I would participate if I was diagnosed with an incurable disease and clinical trial could help me to extend my life.*" Another important element brought up by one participant was that there need to be adequate measures taken to guarantee that the ethical values are not being violated. It was put as: "*. . .if sufficient reassurance that all measures have been taken to guarantee the physical, moral and intellectual safety of the participants in the clinical trial, I think I may participate.*"

**Table 4. Assessing perception among those who heard about clinical trials.**

| | Statements related to perception | Percentage of respondents that agree with the statements | | |
|---|---|---|---|---|
| | | Did not participate in a CT before (N = 57) | Participated in a CT before (N = 126) | Total (N = 183) |
| V1 | Clinical trials being conducted are beneficial to the society. | 93.7% | 98.2% | 95.1% |
| V2 | Clinical research being conducted is harmful to the society. | 93.7% | 86.0% | 91.3% |
| V3 | Clinical research is an essential step in developing new treatment and medical devices. | 79.4% | 87.7% | 83.5% |
| V4 | It is essential to conduct experiment on human beings for the advancement of science. | 88.9% | 89.5% | 89.1% |
| M1 | People participate in clinical trial because it helps to develop new drugs and treatments. | 71.4% | 89.5% | 77.0% |
| M2 | People participate in clinical research mostly for financial gain. | 88.9% | 89.5% | 89.1% |
| M3 | Participation in research is entirely voluntary. | 79.4% | 87.7% | 82.0% |
| M4 | People participate in clinical trials because they want to help the society. | 71.4% | 89.5% | 77.0% |
| C1 | Volunteers are sufficiently compensated while participating in clinical trials. | 80.2% | 82.5% | 80.9% |
| C2 | Confidentiality is respected for volunteers participating in clinical trials. | 62.7% | 66.7% | 63.9% |
| C3 | Volunteers in clinical trials are properly informed about the research they participate in. | 47.6% | 61.4% | 51.9% |
| C4 | Harmful events occurring during a clinical trial must be due to experimental treatment. | 71.4% | 78.9% | 73.8% |
| C5 | The public should be involved in clinical research (e.g., design, oversight, funding). | 71.4% | 77.2% | 73.2% |
| C6 | Researchers make sure research is safe for participants. | 59.5% | 59.6% | 59.6% |
| T1 | The government satisfactorily protects the public against unethical clinical trials being conducted. | 63.5% | 66.7% | 64.5% |
| T2 | Information provided by pharmaceutical companies regarding clinical trials can be trusted. | 67.5% | 77.2% | 70.5% |
| T3 | Doctors force their patients to participate in research. | 52.4% | 45.6% | 50.3% |
| T4 | Human participants in clinical research are treated like experimental animals ('human Guinea Pigs'). | 60.3% | 56.1% | 59.0% |
| T5 | Confidentiality is a matter of importance to research participant. | 42.9% | 40.4% | 42.1% |
| T6 | The media accurately describes clinical research. | 54.0% | 36.8% | 48.7% |
| | Overall perception | 65.7% | 76.9% | |

CT: clinical trials.

V1-4: assessing perception related to value and importance of clinical trials.

M1-4: Assessing perception related to motivating factors behind participation in clinical trials.

C1-4: Assessing perception related to conduct of clinical trials.

T1-6: Assessing perception related to trust in government and pharmaceutical companies while conducting research.

Last but not least, a minority of people interviewed said that only people with serious health problems should participate in clinical trials. Healthy people should not participate in clinical trials since they may bear the consequences of the side effects. For example, one participant said, "*I think opportunity to participate in clinical trials should be given to people with serious health conditions.*" Quantitatively, 62% people were willing to participate in a trial if they had all the required information.

## New cure and altruism

Although, there was a general perception that clinical trials are for patients with chronic illnesses, a degree of altruism was prevalent. Participants thought that clinical trials would help

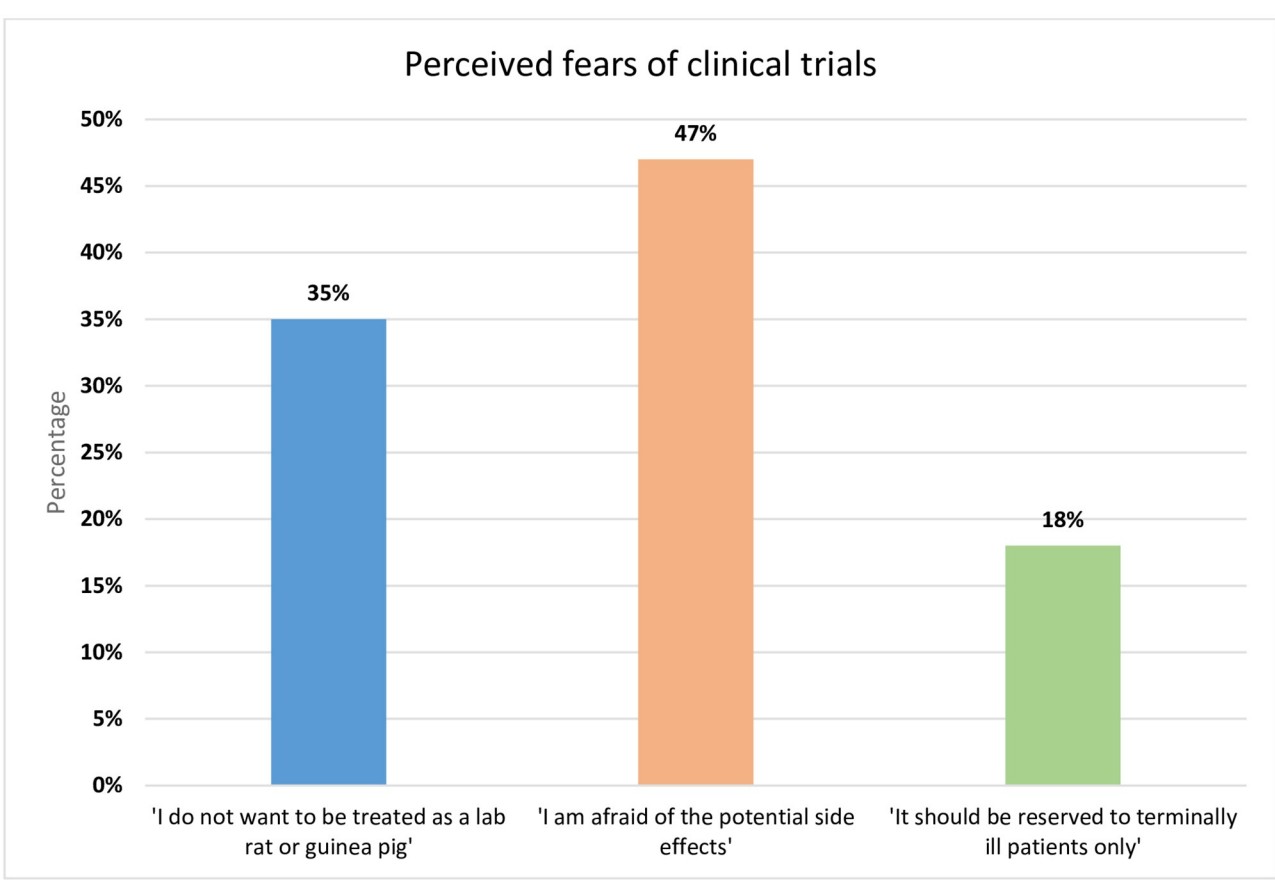

**Fig 1. Perceived fears of clinical trials.**

the society by finding new ways and drugs to cure people suffering from various kinds of illnesses. One participant said: "*More and better treatment options are discovered. They are bound to be more efficient and beneficial for patient.*", while another participant stated: "*Yes. If it can help to find a cure for certain diseases and help the humanity, I would participate.*" Participants also mentioned the prospective idea of being able to help close relatives suffering from a particular disease and also helping future generations as a positive outcome of clinical trials. One participant said: "*I think that clinical trials can help our family, and people*".

## Perceived benefits and fears of clinical trials

While fear and poor perception were highly noted among participants in the qualitative part, around 47% of participants were afraid of potential side effects, while 35% claimed they do not want to be treated as 'guinea pigs'. Participants strongly associate clinical trials to tests on animals and they fear being 'sacrificed' post trial (Fig 1). Participants further strongly perceived remuneration (41%) as one of the driving factors to engage in a clinical trial (Fig 2).

## Discussion

This mixed methods study carried out in an emerging economic setting shed some lights on pertinent issues concerning the perception of clinical trials.

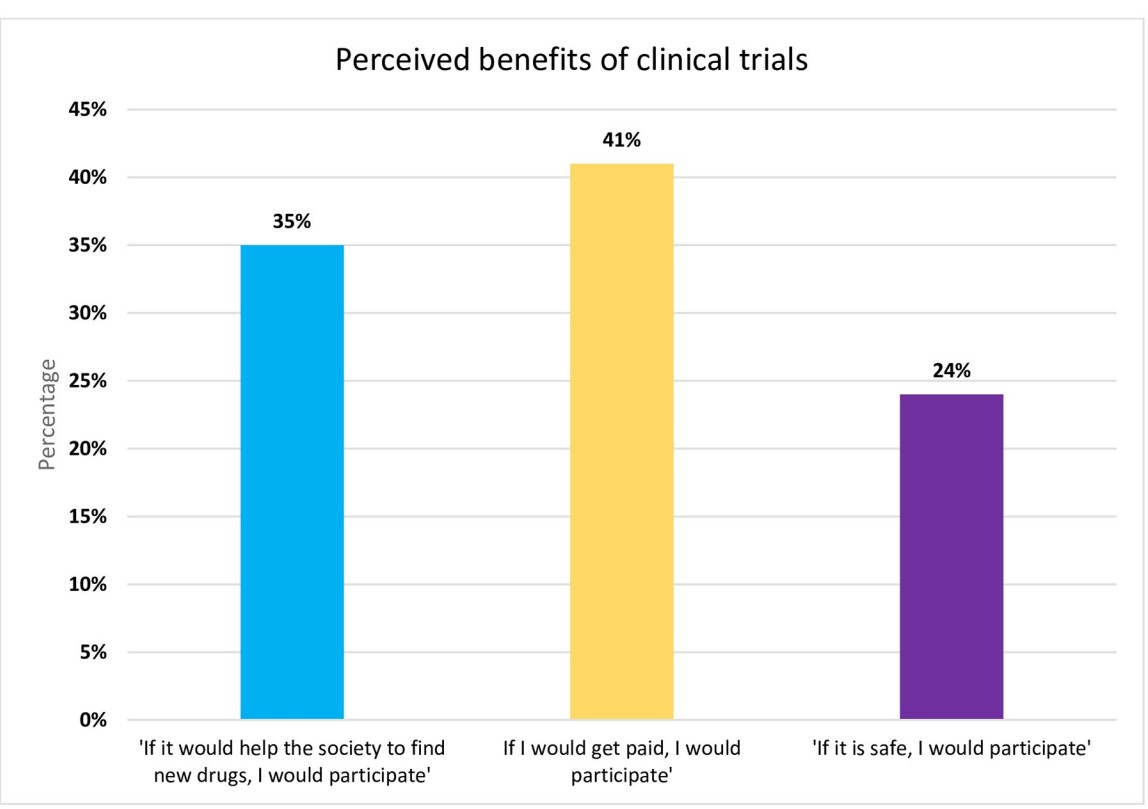

**Fig 2. Perceived benefits of clinical trials.**

### Awareness and knowledge of clinical trials

Our mixed methods study demonstrated that 47.8% of the participants did not know about the existence of clinical trials. Studies conducted in relation to participant recruitment demonstrated that inadequate understanding of clinical trials among the population is a real hindrance to engaging participants in them [3, 10]. Similar conclusion was reached by a study conducted in India in 2016 which linked ignorance of clinical research among the general population to barriers in recruitment and running of clinical trials [11]. On the other side, a lack of awareness of clinical trials among the general Mauritian can lead to over exploitation of the naïve population. Clinical research conducted in Low to Moderate Income Countries (LMIC) often raises concern about this particular issue [12, 13]. Moreover, the establishment of the Clinical Trial Act has now opened the doors to many pharmaceutical companies to conduct research in the island. It is thus imperative that Mauritians are aware of their rights and they are not subject to over exploitation. A mounting interest to conduct research in LMICs demands for a balance to be found between providing health benefits and over exploitation of the economically disadvantaged population [14]. There are numerous reports written over the years to protect participants in clinical trials especially for those who come from LMIC [15, 16].

While a study conducted by Leiter et al. in 2015 found no association between age and clinical trial awareness, our study showed that respondents who knew about clinical trials were slightly younger than those who did not know about clinical trials [17]. This finding has a key implication and can be useful to stakeholders when designing target specific programs where they can sensitize and raise awareness of clinical research among the elderly population.

Similarly, Cameron et al. in 2013 suggested the use of educational programs to improve the knowledge of the elderly population on clinical trials [18]. People who knew about clinical trials in our study had a higher level of education compared to those who did not know about clinical trials. Leiter A et al. showed that educational level was closely related to awareness of clinical research. A low level of education was shown to have an association with unawareness of clinical trials [17]. This association was also found in a study conducted by Lanford A et al. [19]. In the Public Awareness of Research for Therapeutic Advancements through Knowledge and Empowerment (PARTAKE) study most of respondents were housewives and reported that they did not know about clinical trials [6]. In our study however we found that most people who were unemployed expressed that they did not know about clinical trials (44.3%). Some studies conducted to investigate the relationship between level of income and clinical trial awareness found that people with low income were significantly less likely to have heard of clinical research [17, 20]. However, our study did not find a statistically significant relationship between income level and clinical trial awareness. No notable difference between these 2 variables was also obtained in a study conducted by Choi et al. in 2016 [21].

Overall, 56.8% respondents had poor knowledge of clinical trials. Poor knowledge prevailed more in people who never participated in a clinical trial before (42.1%). A study conducted by Cameron et al. in 2013 highlighted the fact that poor knowledge of clinical trials often leads to poor participant recruitment [18]. Although the methodology and the scoring system of assessing knowledge differed, a study conducted by Brandberg et al. in 2016 concluded that there were positive associations between knowledge and participation [22]. We can thus postulate that by educating people about clinical research, the barriers to engaging the public will be less consequent. However, in depth analysis needs to be undertaken to tackle this aspect of clinical research.

## Perception of clinical trials

Globally, a good perception was noted amongst those who knew about clinical trials (N = 183) regarding the value of clinical research. Planned comparisons between those who participated and those who did not participate in a clinical trial did not reveal any statistical difference in regards to value of clinical trial (p value >0.050). Approximately 95.1% respondents agreed with the statement '*clinical research benefits mankind*' and 89.1% disagreed with the statement '*clinical research harms the society*'. This key finding in our study ties well with previous studies conducted in the same field. A study conducted over 68 countries in 2018 showed similar findings [23]. Two other studies conducted in India and Korea respectively showed that the respondents endorsed the main aim goals and benefits of clinical trials [2, 6]. Based on the findings of previous studies and ours, it is seemingly safe to say that general population across many countries believes that clinical trials are beneficial to mankind.

A good proportion of respondents in our study perceived advancement of science to be the most important reason to develop new drugs (77.0%). In line with this finding, a study conducted by Anderson et al. in 2018 showed that 84.5% of the 10 506 individuals surveyed perceived clinical research to be very important in the advancement of science [23].

Even though a high proportion of participants expressed positive perceptions about the conduct of clinical research in Mauritius, 19.1% of participants in our study perceived that volunteers do not obtain adequate information and 20.2% respondents did not believe that confidentiality is adequately protected. Even though it is a minority that has expressed concerns these pertinent issues, the latter needs to be tackled carefully. While it is not uncommon for underprivileged societies who have scarce resources and poor access to healthcare to indulge in accepting the burden of participation in clinical research more than the privileged ones, it is

crucial to ensure that ethical requirements which are in line with the Belmont report and Declaration of Helsinki are thoroughly respected [14, 24].

Trust in pharmaceutical companies, the government and ethical issues pertaining to confidentiality were perceived poorly in our study. Around 32.8% of the respondents did not agree with the statement that the government adequately protects the public against unethical clinical research and 28.4% reported that pharmaceutical companies cannot be trusted. Similar findings have been found in the PARTAKE study where there was a minority of participants who had negative perceptions regarding trust in research companies [6]. It is thus crucial to highlight the essence of having transparency in the conduct of clinical trials to provide better outcome and thus increasing the trust of participants [25].

## Perception and previous participation in clinical trials

While in our study we found that 31.1% of our respondents who heard about clinical trials had previously participated in one, we found that the average percentage of interviewees that had previously participated in other research was much lower in other countries. For instance, the PARTAKE study carried out in India revealed that 2.9% of their respondents had previously participated in a clinical trial [6]. Additionally a study conducted in Europe revealed that the rate was 6% [26]. One plausible explanation to account for our high rate of experienced participants compared to other studies can be due to supermarkets being in close proximity to the of clinical trial centers and also most of the clinical trial centers come from the Plaine Wilhems district, which is the biggest source of our participants.

Overall, respondents who participated in a clinical trial had a better perception of the latter compared to those who did not participate in one. This finding is in line with The Center for Information and Study on Clinical Research Participation (CISCRP) study conducted in 2017 which concluded that previous participation indeed improves perception [4]. Likewise, a study conducted in Mexico in 2016 found that 90% out of 604 previous participants of clinical trials would like to enroll in another clinical trial again [27]. However, there were 2 concerning findings that came to light in our study. Firstly, 45.6% of respondents who previously participated in clinical trials agreed with the statement T3: '*Doctors force their patients to participate in research*'. Secondly, 38.6% of respondents who previously participated in a clinical trial did not agree with the statement C3: '*Volunteers in clinical trials are properly informed about the research they participate in*'. These 2 findings raise concerns as to whether there has been coercion by physicians and inadequate information were being provided to them while enrolling in clinical research. This lays ground for further studies to be carried out to investigate whether participants are being adequately informed by the pharmaceutical companies and doctors. This theme was beyond the scope of our study. A valid and reliable instrument to measure perceived coercion among participants enrolled in a clinical trial can be developed for further investigation.

## Level of education and perception of clinical trials

A high level of education was associated with a better perception of clinical trials among Mauritians. An association between high level of education and good perception has also been observed in a study conducted by Chu et al. in 2015 [3]. An important implication of this finding would be that when designing awareness campaigns on clinical trials, specific populations where low educational prevails can be targeted.

## Perceived fears and benefits of clinical trials

47% of participants were afraid of the potential side effects of intervention related to the clinical research. The results obtained from our study are in line with various studies conducted on

recruitment barriers in clinical trials across the world that showed fear of adverse effect from unknown drugs as a strong barrier to participation [2, 3, 28]. Approximately 35% of people from our study stated that an important deterrent factor from participating in a clinical trial is the fear of being treated as a "guinea pig'. The latter is a metaphor that participants in many studies conducted across the world have expressed concerns about and it was often related to mistreatment and exploitation [24, 29, 30].

Up to 76% of our respondents perceived the national and local benefits one might obtain from clinical trials being conducted in the island. A study carried out by Weigmann et al. in 2015 supported this belief and concluded that conducting clinical trials in LMIC can be beneficial to the community, since it not only provides the people with affordable drugs to locally prevalent diseases but can also help to boost the economy and consolidate health care facilities [15]. It is crucial to point out that unfortunately even though the burden of health is higher in LMIC, statistics show that there is a significant under representation of LMIC in clinical trials on a global platform [31].

This mixed study revealed that compensation was a factor to favor participation in clinical research. This was further endorsed quantitatively as 41% population stated that they would participate if they were remunerated. While a study carried out by Walsh and Sheridan in 2016 did show that money was a dominant factor to participate in research, [10] another study conducted by Stocks et al. in 2012 which used money as an incentive to increase attendance for cardiovascular risk assessment revealed that money as an incentive did not influence participation in clinical trials [32]. Holman et al. [20] who investigated the factors affecting participation in Randomized Clinical Trials (RCT) for patients with fibromyalgia identified that that although incentives were considered as an important factor determining participation, they were considered to be least influential [20].

Willison et al. (2018) showed that respondents were more inclined to identify societal benefits of clinical trials over personal benefits [7]. In support of this belief, our qualitative study noted some altruistic motives in favor of participation which was further strengthened when 35% of participants in our quantitative study said that they would participate if it would benefit the society.

## Strengths and limitations of the study

Our mixed methods study explored an important aspect of clinical research in the middle income setting. It identified key aspects pertaining to awareness, perception and knowledge of clinical trials which can be important learning needs for the general population as well as investigators. The use of a mixed methods study to analyze the perception of clinical trials among Mauritian citizens allowed us to develop a psychometric instrument well adapted to the Mauritian population. Thus, we were able to have a comprehensive understanding of the research being carried out. Our study encompassed participants from various ethnicities, socioeconomic status and income levels. The perception and knowledge that were assessed were thus broadly representative of the perception of the Mauritian population with regards to clinical research.

A minimum acceptable sample size was used for this study due to time constraint. Hence we suggest that larger cross-sectional surveys need to be carried out to evaluate perception of clinical trials on a wider scale. Our mean age was 39.9 ±14.6 which represents a rather young population. The perception and awareness of older people with co-morbidities were hence not truly appreciated in this study. This is mainly due to our study design which relied on people who went to the supermarkets and therefore we might have excluded those who are very old or have physical limitations.

On retrospection, an association between willingness to participate among those who heard about clinical trials could not be assessed. This leaves ground for future scope for research.

## Implications of this study and recommendations

A lack of information and awareness observed in this mixed methods study makes it imperative to have awareness campaigns and educational programs to enhance public engagement in clinical research and to make them aware of their rights as potential participants. Collaborative efforts of various stakeholders in clinical trials should ensure that effective means to disseminate accurate and comprehensive information are being deployed in a population centered manner targeting especially those with low literacy level and the elderly population. Secondly, poor perception regarding trust and conduct in clinical trials can be tackled by engaging the population in a two-way communication where the general population become active partners in the research process. We advise in the frequent use of surveys and questionnaires to evaluate the perception, attitudes and level of satisfaction of participants enrolled in clinical trials. A close comprehension on the beliefs, experiences and perception of participants enrolled in clinical trials can lead to an effective and continuous enhancement in patient engagement. Finally, we suggest policymakers and the government to allocate grants for non-profit organizations whose main aim will be to focus on the health of the population and instill confidence in them if they are considering clinical trial as a care option.

## Ethical implications

Conduction of clinical research in a naïve and vulnerable population such as the Mauritian population should be done under strict regulations to ensure that human rights are not being violated and clinical research being conducted are stringent and according to guidelines set by the Declaration of Helsinki. Physicians should ensure that information given to participants is free from any biased representation or coercion. Informed consent which is a vital aspect of research should be tailor made to the Mauritian context and translated to the native Mauritian language to ensure that the participants have diligently understood everything.

## Conclusion

This study highlighted crucial points pertaining to perception of clinical trials. It also provided a foundation to guide future works. Target specific awareness campaigns and educational programs can be undertaken by stakeholders to engage and promote clinical trials among the general Mauritian population. Further more in-depth and larger cross sectional surveys need to be carried out to assess the predictors that influence perception of clinical trials.

## Acknowledgments

A huge thanks to all participants who heartily participated in this study.

## Author Contributions

**Conceptualization:** Wafaa Binti Mowlabaccus, Abha Jodheea-Jutton.

**Data curation:** Wafaa Binti Mowlabaccus.

**Formal analysis:** Wafaa Binti Mowlabaccus.

**Investigation:** Wafaa Binti Mowlabaccus, Abha Jodheea-Jutton.

**Methodology:** Wafaa Binti Mowlabaccus, Abha Jodheea-Jutton.

**Project administration:** Abha Jodheea-Jutton.

**Resources:** Wafaa Binti Mowlabaccus.

**Software:** Wafaa Binti Mowlabaccus.

**Supervision:** Abha Jodheea-Jutton.

**Validation:** Wafaa Binti Mowlabaccus.

**Visualization:** Wafaa Binti Mowlabaccus, Abha Jodheea-Jutton.

**Writing – original draft:** Wafaa Binti Mowlabaccus, Abha Jodheea-Jutton.

**Writing – review & editing:** Wafaa Binti Mowlabaccus, Abha Jodheea-Jutton.

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
