## [Decision Letter · Decision Letter 0]

15 Jun 2020

PONE-D-20-13417

Participant perception, still a major challenge to clinical research in developing countries- A mixed methods study

PLOS ONE

Dear Dr. Mowlabaccus,

Thank you for submitting your manuscript to PLOS ONE. After careful consideration, we feel that it has merit but does not fully meet PLOS ONE’s publication criteria as it currently stands. Therefore, we invite you to submit a revised version of the manuscript that addresses the points raised during the review process.

ACADEMIC EDITOR:

Congratulations! The reviewers thought that the submission has merit for publication, and only had minor comments. If you could address them, we can move forward with publication. Thanks for considering us for your scientific work.

We look forward to receiving your revised manuscript.

Kind regards,

Zhi Ven Fong, M.D., M.P.H.

Academic Editor

PLOS ONE

Journal Requirements:

2. Please address the following:

- Please include additional information regarding the interview guide and questionnaire used in the study and ensure that you have provided sufficient details that others could replicate the analyses. For instance, if you developed a questionnaire as part of this study and it is not under a copyright more restrictive than CC-BY, please include a copy, in both the original language and English, as Supporting Information.

- Please refrain from stating p values as 0.000, either report the exact value or employ the format p<0.001.

3.  Thank you for including your ethics statement:

"The ethical clearance was granted by the Department of Medicine Research ethics committee on the 23rd of August 2018. Consent was obtained in written and all the data were analyzed anonymously.".

i) Please amend your current ethics statement to include the full name of the ethics committee/institutional review board(s) that approved your specific study.

ii) Once you have amended this/these statement(s) in the Methods section of the manuscript, please add the same text to the “Ethics Statement” field of the submission form (via “Edit Submission”).

4. Please ensure that you refer to Figure 1 in your text as, if accepted, production will need this reference to link the reader to the figure.

5. Please upload a copy of Figure 3, to which you refer in your text on page 27. If the figure is no longer to be included as part of the submission please remove all reference to it within the text.

Reviewers' comments:

Reviewer's Responses to Questions

**Comments to the Author**

1. Is the manuscript technically sound, and do the data support the conclusions?

Reviewer #1: Yes

Reviewer #2: Yes

2. Has the statistical analysis been performed appropriately and rigorously? 

Reviewer #1: Yes

Reviewer #2: Yes

3. Have the authors made all data underlying the findings in their manuscript fully available?

Reviewer #1: Yes

Reviewer #2: Yes

4. Is the manuscript presented in an intelligible fashion and written in standard English?

Reviewer #1: Yes

Reviewer #2: Yes

5. Review Comments to the Author

Reviewer #1: First off, I would like to congratulate the authors on a well concieved and excecuted project. I especially liked the use of a mixed methods format to answer this question. The manuscript is well written, organized and easy to follow. The tables and figures are generally easy to interpret and add to the presentation of the data.

I have only a few minor suggestions that the authors can consider:

1. Regarding table 1: It might be helpful to add a footnote about what the different income quintiles mean. For example, which of these is/are below the poverty line. Most of the readers will be unfamiliar with the currency.

2. Page 7 says "around 23" people filled out the semi structured form. Is it "around" 23? This number sounds reasonably exact.

3. What was the recruitment and response rate for both parts of this study. It would be helpful to know how many people were approached? How many agreed to participate? How many were included in the final analysis?

4. Why was the age limit 20-65 in part 1 and 20-70 in part 2? Why is there an upper limit for age at all? You might also mention that your study design relied on people who went to the supermarket and therefore might exclude those who are very old or have physical limitations.

5. It seems amazing that 57 of the 183 people that knew about CT's had previously participated in a clinical trial. Is this rate of prior participation higher than in other studies? How do you account for the fact that so many had personally participated in CT's before?

6. I think some of the findings from people who had previously participated in CT's were interesting and probably deserve comment. Is it true, for instance, that when you asked (T3) "do doctors force patients to participate" that the majority of people who had previously participated in CT's before got this wrong? Does that mean that they previously felt coerced? Or for question (C3) "volunteers are properly informed" did 40% of people who had previously participated in clinical trials get this wrong? Does that mean they didn't feel they were informed in the prior CT they participated in?

7. Table 4 is a bit misleading. You state the percentages represent the percent of "correctly answered questions." The table is not labeled as such. I think you have 2 options: You can keep the table as it is but make the labeling more clear or you can change the percentages from "percent correct" to "percent that agree" with these statements. I make this distinction because some of these questions do not appear to have a "correct" answers. For instance, "The media accurately describes clinical research" - it is a bit subjective, what is the "correct" answer to this? Or "people participate in clinical trials because they want to help society." This is the motivation for some, but not for all, what is the "correct" answer here?

Reviewer #2: Overall, I thought that this paper was well thought out and executed. The authors are very clear in their intent and have put a lot of work into the design and implementation of their mixed methods approach. This includes a method for extracting themes in the qualitative arm and ensuring adequate power in the quantitative arm. And the quantitative portion of the analysis is in line and supportive of the findings.

Furthermore, the findings pose an interesting question regarding expansion of investigative trials into economically developing, or otherwise disadvantaged, settings. On the one hand these areas have a population that has not been exposed to many of the influences and interventions that other more mature economies have seen and perhaps you can assess true physiology “nature”; on the other hand if an intervention has a larger “nurture” component then this would be missed, one may question generalizability if the population is different in composition than a population elsewhere in the world. Even more interestingly there is the moral/ethical question of subjecting these persons to interventions. The authors touch on this a bit in the discussion, and I feel that this adds an important timely and topical aspect to the manuscript. If anything, this could be expanded somewhat to consider moral/ethical questions that arise at different levels: certainly the drug companies as mentioned in the manuscript, but one can also discuss these issues at a government/revenue level, or at a personal level when deciding whether to enroll yourself as a sole breadwinner or a dependent family member.

As a final comment, while the methodology and discussion is interesting, I feel that the paper itself could benefit from re-review to fix the stylistic or spelling errors, some of which are noted below.

Comments:

1. Page 2. Consider an alternative acronym for clinical trials. “CT” is so commonly used to represent computed tomography imaging, that this can be a bit confusing at first. Small point, because it becomes less noticeable over time.

2. Page 2. Technically called a “mixed methods study”. It is correctly designated as such on p5 but not consistently throughout the manuscript.

3. Page 2. “ratified” as a word choice in abstract, more of legal connotation, consider changing to something like “agreed with”

4. Style difference in wording throughout “Up to date” p5; “…who were literate” p6 consider calling them “…literate participants”; “till” p.6; “…among general Mauritian” p.7 should be “…among (or amongst) the general…”; “for the quantitative one” p. 7, perhaps better stated “for the quantitative study”; “…one exceptional precise answer…” p9, should be “…one exceptionally accurate answer” as exception should be an adverb here and precision and accuracy are different concepts with this response being both precise and accurate but in this context its accuracy is what is important; “A notable discrepancy was noted…” should eliminate one instance of the word noted .22;

5. Table 1, right hand column should be called “p value” or something to that effect

6. Would broaden the moral/ethical implications in the discussion section

6. PLOS authors have the option to publish the peer review history of their article (what does this mean?). If published, this will include your full peer review and any attached files.

Reviewer #1: Yes: Geoffrey A Anderson

Reviewer #2: Yes: Sahael Stapleton

---

## [Author Response · Author response to Decision Letter 0]

6 Jul 2020

We have addressed the comments from the journal editor as follows: 

1. We ensured that the manuscript meets PLOS ONE style requirements, including those for file naming. 

2. We included additional supporting information containing the participant information sheet and the questionnaires used as S1 file in p6 para 2. 

3. We reported all values as exact values in 2 decimal places. 

4. We included the full name of our ethics committee and has added the same text to the ethics statement.

5. We referred to Fig 1 in the text.

6. We removed all reference related to figure 3. 

We have responded to the comments of reviewer 1 as follows:

1. Regarding table 1: It might be helpful to add a footnote about what the different income quintiles mean. For example, which of these is/are below the poverty line. Most of the readers will be unfamiliar with the currency.

Thank you for this suggestion. A footnote was added below the table 1(p.9) which included that an income of less than 5000 Mauritian rupees ($124.30 US) per month was considered below the poverty line. We added the amount of US dollar in brackets for people who are unfamiliar with this currency.

2. Page 7 says "around 23" people filled out the semi structured form. Is it "around" 23? This number sounds reasonably exact.

Thank you for pointing that out. In fact, it is 23 people. We have corrected it in p.8 para 1. 

3. What was the recruitment and response rate for both parts of this study. It would be helpful to know how many people were approached? How many agreed to participate? How many were included in the final analysis?

Thank you for asking for more clarification. We have added the response rate for both parts of the mixed study (p.8, para 1). It is as follows: “To get in-depth insights into the perceptions of adults on clinical trials, a total of 35 questionnaires were distributed for the qualitative study and 23 participants filled the semi-structured form. For the quantitative study, 400 questionnaires were distributed of which 364 were collected and 350 of them were considered to be properly filled. The response rate for part I of the study was 67.5% while that of part II was 87.5%.’’ 

4. Why was the age limit 20-65 in part 1 and 20-70 in part 2? Why is there an upper limit for age at all? You might also mention that your study design relied on people who went to the supermarket and therefore might exclude those who are very old or have physical limitations.

Thank you for pointing that out. Actually it was 70 years for both parts. We aimed at this upper limit since we were less likely to get participants beyond 70 years since they rarely go to supermarkets. The age has been corrected for part I (p.6, para 2). We have included your suggestion in our limitations. It is included as ‘’ The perception and awareness of older people with co-morbidities were hence not truly appreciated in this study. This is mainly due to our study design which relied on people who went to the supermarkets and therefore we might have excluded those who are very old or have physical limitations (p.26, para 1)’’.

5. It seems amazing that 57 of the 183 people that knew about CT's had previously participated in a clinical trial. Is this rate of prior participation higher than in other studies? How do your account for the fact that so many had personally participated in CT's before?

Thank you for highlighting this. We have added a new paragraph in the discussion section which highlights this p.22,p23 para2. It is written as follows: “While in our study we found that 31.1 % of our respondents who heard about clinical trials had previously participated in one, we found that the average percentage of interviewees that had previously participated in other studies were much lower. For instance, the PARTAKE study carried out in India revealed that 2.9% of their respondents had previously participated in a clinical trial [7]. Additionally a study conducted in Europe revealed that the rate was 6% [26]. One plausible explanation to account for our high rate of experienced participants compared to other studies can be due to supermarkets being in close proximity to the of clinical trial centers and also most of the clinical trial centers come from the Plaine Wilhems district, which is the biggest source of our participants.’’

6. I think some of the findings from people who had previously participated in CT's were interesting and probably deserve comment. Is it true, for instance, that when you asked (T3) "do doctors force patients to participate" that the majority of people who had previously participated in CT's before got this wrong? Does that mean that they previously felt coerced? Or for question (C3) "volunteers are properly informed" did 40% of people who had previously participated in clinical trials get this wrong? Does that mean they didn't feel they were informed in the prior CT they participated in?

Thank you for pointing this out. Indeed, these findings deserve comment. We have added a paragraph in p.23: ‘’ However, there were 2 concerning findings that came to light in our study. Firstly, 45.6 % of respondents who previously participated in clinical trials agreed with the statement T3: ‘’Doctors force their patients to participate in research’’. Secondly, 38.6% of respondents who previously participated in a clinical trial did not agree with the statement C3: ‘Volunteers in clinical trials are properly informed about the research they participate in. These 2 findings raise concern as to whether there has been coercion by physicians and inadequate information were being provided to them while enrolling in clinical research. This lays ground for further studies to be carried out to investigate whether participants are being adequately informed by the pharmaceutical companies and doctors. This theme was beyond the scope of our study. A valid and reliable instrument to measure perceived coercion among participants enrolled in a clinical trial can be developed for further investigation.’’

7. Table 4 is a bit misleading. You state the percentages represent the percent of "correctly answered questions." The table is not labeled as such. I think you have 2 options: You can keep the table as it is but make the labeling more clear or you can change the percentages from "percent correct" to "percent that agree" with these statements. I make this distinction because some of these questions do not appear to have a "correct" answers. For instance, "The media accurately describes clinical research" - it is a bit subjective, what is the "correct" answer to this? Or "people participate in clinical trials because they want to help society." This is the motivation for some, but not for all, what is the "correct" answer here?

We share your concerns. We have added a new row at the top of the table 4 which now has a clearer labelling which read as “Percentage of respondents that agree with the statements” (p.14). 

We have responded to the comments from reviewer 2 as follows:

1. Page 2. Consider an alternative acronym for clinical trials. “CT” is so commonly used to represent computed tomography imaging, that this can be a bit confusing at first. Small point, because it becomes less noticeable over time.

We share your concerns and since we have been unable to find a suitable acronym other than CT, we have decided to change all ‘’CT’’ in the manuscript to clinical trials or clinical research to avoid any confusion as stated above. 

2. Page 2. Technically called a “mixed methods study”. It is correctly designated as such on p5 but not consistently throughout the manuscript.

Thank you for pointing this out. We have now mentioned it throughout the manuscript. 

3. Page 2. “ratified” as a word choice in abstract, more of legal connotation, consider changing to something like “agreed with”

Thank you for suggesting an alternative word. We have changed it.

4. Style difference in wording throughout “Up to date” p5; “…who were literate” p6 consider calling them “…literate participants”; “till” p.6; “…among general Mauritian” p.7 should be “…among (or amongst) the general…”; “for the quantitative one” p. 7, perhaps better stated “for the quantitative study”; “…one exceptional precise answer…” p9, should be “…one exceptionally accurate answer” as exception should be an adverb here and precision and accuracy are different concepts with this response being both precise and accurate but in this context its accuracy is what is important; “A notable discrepancy was noted…” should eliminate one instance of the word noted .22;

We have taken into consideration all the style difference in wordings you have suggested and modified it in our manuscript accordingly. 

5. Table 1, right hand column should be called “p value” or something to that effect

Thank you for suggesting this. The right hand column in table 1 is now called the p value. 

6. Would broaden the moral/ethical implications in the discussion section

Thank you for your suggestion. We have added a new paragraph on the ethical implications (p.28)

‘’Conduction of clinical research in a naïve and vulnerable population such as the Mauritian population should be done under strict regulations to ensure that human rights are not being violated and clinical research being conducted are stringent and according to guidelines set by the Declaration of Helsinki. Physicians should ensure that information given to participants is free from any biased representation or coercion. Informed consent which is a vital aspect of research should be tailor made to the Mauritian context and translated to the native Mauritian language to ensure that the participants have diligently understood everything. ‘’

---

## [Editor Report · Decision Letter 1]

10 Jul 2020

Participant perception, still a major challenge to clinical research in developing countries- A mixed methods study

PONE-D-20-13417R1

Dear Dr. Mowlabaccus,

We’re pleased to inform you that your manuscript has been judged scientifically suitable for publication and will be formally accepted for publication once it meets all outstanding technical requirements.

Kind regards,

Zhi Ven Fong, M.D., M.P.H.

Academic Editor

PLOS ONE
---

## [Editor Report · Acceptance letter]

17 Jul 2020

PONE-D-20-13417R1 

Participant perception, still a major challenge to clinical research in developing countries- A mixed methods study 

Dear Dr. Mowlabaccus:

I'm pleased to inform you that your manuscript has been deemed suitable for publication in PLOS ONE. Congratulations! Your manuscript is now with our production department. 

Kind regards, 

on behalf of

Dr. Zhi Ven Fong 

Academic Editor

PLOS ONE